# An Interpretable Knowledge Transfer Model
# for Knowledge Base Completion

## Abstract

Knowledge bases are important resources for a variety of natural language processing tasks but suffer from incompleteness. We propose a novel embedding model, *ITransF*, to perform knowledge base completion. Equipped with a sparse attention mechanism, ITransF discovers hidden concepts of relations and transfer statistical strength through the sharing of concepts. Moreover, the learned associations between relations and concepts, which are represented by sparse attention vectors, can be interpreted easily. We evaluate ITransF on two benchmark datasets—WN18 and FB15k for knowledge base completion and obtains improvements on both the mean rank and Hits@10 metrics, over all baselines that do not use additional information.

## 1 Introduction

Knowledge bases (KB), such as WordNet (Fellbaum, 1998), Freebase (Bollacker et al., 2008), YAGO (Suchanek et al., 2007) and DBpedia (Lehmann et al., 2015), are useful resources for many applications such as question answering (Berant et al., 2013; Yih et al., 2015; Dai et al., 2016) and information extraction (Mintz et al., 2009). However, knowledge bases suffer from incompleteness despite their formidable sizes (Socher et al., 2013; West et al., 2014), leading to a number of studies on automatic knowledge base completion (KBC) (Nickel et al., 2015) or link prediction.

The fundamental motivation behind these studies is that there exist some statistical regularities under the intertwined facts stored in the multi-relational knowledge base. By discovering generalizable regularities in known facts, missing ones may be recovered in a faithful way. Due to its excellent generalization capability, distributed representations, a.k.a. embeddings, have been popularized to address the KBC task (Nickel et al., 2011; Bordes et al., 2011, 2014, 2013; Socher et al., 2013; Wang et al., 2014; Guu et al., 2015; Nguyen et al., 2016b).

As a seminal work, Bordes et al. (2013) proposes the TransE, which models the statistical regularities with linear translations between entity embeddings operated by a relation embedding. Implicitly, TransE assumes both entity embeddings and relation embeddings dwell in the same vector space, posing an unnecessarily strong prior. To relax this requirement, a variety of models first project the entity embeddings to a relation-dependent space (Bordes et al., 2014; Ji et al., 2015; Lin et al., 2015b; Nguyen et al., 2016b), and then model the translation property in the projected space. Typically, these relation-dependent spaces are characterized by the projection matrices unique to each relation. As a benefit, different aspects of the same entity can be temporarily emphasized or depressed as an effect of the projection. For instance, STransE (Nguyen et al., 2016b) utilizes two projection matrices per relation, one for the head entity and the other for the tail entity.

Despite the superior performance of STransE compared to TransE, it is more prone to the data sparsity problem. Concretely, since the projection spaces are unique to each relation, projection matrices associated with rare relations can only be exposed to very few facts during training, resulting in poor generalization. For common relations, a similar issue exists. Without any restrictions on the number of projection matrices, logically related or conceptually similar relations may have distinct projection spaces, hindering the discovery, sharing, and generalization of statistical regularities.

Previously, a line of research makes use of external information such as textual relations from web-scale corpus or node features (Toutanova et al., 2015; Toutanova and Chen, 2015; Nguyen et al., 2016a), alleviating the sparsity problem. In parallel, recent work has proposed to model regularities beyond local facts by considering multi-relation paths (García-Durán et al., 2015; Lin et al., 2015a; Shen et al., 2016). Since the number of paths grows exponentially with its length, as a side effect, path-based models enjoy much more training cases, suffering less from the problem.

In this paper, we propose an interpretable knowledge transfer model (ITransF), which encourages the sharing of statistic regularities between the projection matrices of relations and alleviates the data sparsity problem. At the core of ITransF is a sparse attention mechanism, which learns to compose shared concept matrices into relation-specific projection matrices, leading to a better generalization property. Without any external resources, ITransF improves mean rank and Hits@10 on two benchmark datasets, over all previous approaches of the same kind. In addition, the parameter sharing is clearly indicated by the learned sparse attention vectors, enabling us to interpret how knowledge transfer is carried out. To induce the desired sparsity during optimization, we further introduce a block iterative optimization algorithm.

In summary, the contributions of this work are: (i) proposing a novel knowledge embedding model which enables knowledge transfer by learning to discover shared regularities; (ii) introducing a learning algorithm to directly optimize a sparse representation from which the knowledge transferring procedure is interpretable; (iii) showing the effectiveness of our model by outperforming baselines on two benchmark datasets for knowledge base completion task.

## 2 Notation and Previous Models

Let $E$ denote the set of entities and $R$ denote the set of relations. In knowledge base completion, given a training set $P$ of triples $(h, r, t)$ where $h, t \in E$ are the head and tail entities having a relation $r \in R$, e.g., (*Steve Jobs*, `FounderOf`, *Apple*), we want to predict missing facts such as (*Steve Jobs*, `Profession`, *Businessperson*).

Most of the embedding models for knowledge base completion define an energy function $f_r(h, t)$ according to the fact's plausibility (Bordes et al., 2011, 2014, 2013; Socher et al., 2013; Wang et al., 2014; Yang et al., 2015; Guu et al., 2015; Nguyen et al., 2016b). The models are learned to minimize energy $f_r(h, t)$ of a plausible triple $(h, r, t)$ and to maximize energy $f_r(h', t')$ of an implausible triple $(h', r, t')$.

Motivated by the linear translation phenomenon observed in well trained word embeddings (Mikolov et al., 2013), TransE (Bordes et al., 2013) represents the head entity $h$, the relation $r$ and the tail entity $t$ with vectors $\mathbf{h}, \mathbf{r}$ and $\mathbf{t} \in \mathbb{R}^n$ respectively, which were trained so that $\mathbf{h} + \mathbf{r} \approx \mathbf{t}$. They define the energy function as

$$f_r(h, t) = \|\mathbf{h} + \mathbf{r} - \mathbf{t}\|_{\ell}$$

where $\ell = 1$ or $2$, which means either the $\ell_1$ or the $\ell_2$ norm of the vector $\mathbf{h} + \mathbf{r} - \mathbf{t}$ will be used depending on the performance on the validation set.

To better model relation-specific aspects of the same entity, TransR (Lin et al., 2015b) uses projection matrices and projects the head entity and the tail entity to a relation-dependent space. STransE (Nguyen et al., 2016b) extends TransR by employing different matrices for mapping the head and the tail entity. The energy function is

$$f_r(h, t) = \|\mathbf{W}_{r,1}\mathbf{h} + \mathbf{r} - \mathbf{W}_{r,2}\mathbf{t}\|_{\ell}$$

However, not all relations have abundant data to estimate the relation specific matrices as most of the training samples are associated with only a few relations, leading to the data sparsity problem for rare relations.

## 3 Interpretable Knowledge Transfer

### 3.1 Model

As discussed above, a fundamental weakness in TransR and STransE is that they equip each relation with a set of unique projection matrices, which not only introduces more parameters but also hinders knowledge sharing. Intuitively, many relations share some concepts with each other, although they are stored as independent symbols in KB. For example, the relation "(somebody) won award for (some work)" and "(somebody) was nominated for (some work)" both describe a person's high-quality work which wins an award or a nomination respectively. This phenomenon suggests that one relation actually represents a collection of real-world concepts, and one concept

can be shared by several relations. Inspired by the existence of such lower-level concepts, instead of defining a unique set of projection matrices for every relation, we can alternatively define a small set of concept projection matrices and then compose them into customized projection matrices. Effectively, the relation-dependent translation space is then reduced to the smaller concept spaces.

However, in general, we do not have prior knowledge about what concepts exist out there and how they are composed to form relations. Therefore, in ITransF, we propose to learn this information simultaneously from data, together with all knowledge embeddings. Following this idea, we first present the model details, then discuss the optimization techniques for training.

**Energy function** Specifically, we stack all the concept projection matrices to a 3-dimensional tensor $\mathbf{D} \in \mathbb{R}^{m \times n \times n}$, where $m$ is the pre-specified number of concept projection matrices and $n$ is the dimensionality of entity embeddings and relation embeddings. We let each relation select the most useful projection matrices from the tensor, where the selection is represented by an attention vector. The energy function of ITransF is defined as:

$$f_r(h, t) = \| \boldsymbol{\alpha}_r^H \cdot \mathbf{D} \cdot \mathbf{h} + \mathbf{r} - \boldsymbol{\alpha}_r^T \cdot \mathbf{D} \cdot \mathbf{t} \|_\ell \quad (1)$$

where $\boldsymbol{\alpha}_r^H, \boldsymbol{\alpha}_r^T \in [0,1]^m$, satisfying $\sum_i \boldsymbol{\alpha}_{r,i}^H = \sum_i \boldsymbol{\alpha}_{r,i}^T = 1$, are normalized attention vectors used to compose all concept projection matrices in $\mathbf{D}$ by a convex combination. It is obvious that STransE can be expressed as a special case of our model when we use $m = 2|R|$ concept matrices and set attention vectors to disjoint one-hot vectors. Hence our model space is a generalization of STransE. Note that we can safely use fewer concept matrices in ITransF and obtain better performance (see section 4.3), though STransE always requires $2|R|$ projection matrices.

We follow previous work to minimize the following hinge loss function:

$$\mathcal{L} = \sum_{\substack{(h,r,t) \sim P, \\ (h',r,t') \sim N}} \left[ \gamma + f_r(h,t) - f_r(h',t') \right]_+ \quad (2)$$

where $P$ is the training set consisting of correct triples, $N$ is the distribution of corrupted triples defined in section 3.3, and $[\cdot]_+ = \max(\cdot, 0)$. Note that we have omitted the dependence of $N$ on $(h, r, t)$ to avoid clutter. We normalize the entity vectors $\mathbf{h}, \mathbf{t}$, and the projected entity vectors

$\boldsymbol{\alpha}_r^H \cdot \mathbf{D} \cdot \mathbf{h}$ and $\boldsymbol{\alpha}_r^T \cdot \mathbf{D} \cdot \mathbf{t}$ to have unit length after each update, which is an effective regularization method that benefits all models.

**Sparse attention vectors** In Eq. (1), we have defined $\boldsymbol{\alpha}_r^H, \boldsymbol{\alpha}_r^T$ to be some normalized vectors used for composition. With a dense attention vector, it is computationally expensive to perform the convex combination of $m$ matrices in each iteration. Moreover, a relation usually does not consist of all existing concepts in practice. Furthermore, when the attention vectors are sparse, it is often easier to interpret their behaviors and understand how concepts are shared by different relations.

Motivated by these potential benefits, we further hope to learn sparse attention vectors in ITransF. However, directly posing $\ell_1$ regularization (Tibshirani, 1996) on the attention vectors fails to produce sparse representations in our preliminary experiment, which motivates us to enforce $\ell_0$ constraints on $\boldsymbol{\alpha}_r^T, \boldsymbol{\alpha}_r^H$.

In order to satisfy both the normalization condition and the $\ell_0$ constraints, we reparameterize the attention vectors in the following way:

$$\boldsymbol{\alpha}_r^H = \mathrm{SparseSoftmax}(\mathbf{v}_r^H, \mathbf{I}_r^H)$$
$$\boldsymbol{\alpha}_r^T = \mathrm{SparseSoftmax}(\mathbf{v}_r^T, \mathbf{I}_r^T)$$

where $\mathbf{v}_r^H, \mathbf{v}_r^T \in \mathbb{R}^m$ are the pre-softmax scores, $\mathbf{I}_r^H, \mathbf{I}_r^T \in \{0,1\}^m$ are the sparse assignment vectors, indicating the non-zero entries of attention vectors, and the SparseSoftmax is defined as

$$\mathrm{SparseSoftmax}(\mathbf{v}, \mathbf{I})_i = \frac{\exp(\mathbf{v}_i / \tau) \mathbf{I}_i}{\sum_j \exp(\mathbf{v}_j / \tau) \mathbf{I}_j}$$

with $\tau$ being the temperature of Softmax.

With this reparameterization, $\mathbf{v}_r^H, \mathbf{v}_r^T$ and $\mathbf{I}_r^H, \mathbf{I}_r^T$ replace $\boldsymbol{\alpha}_r^T, \boldsymbol{\alpha}_r^H$ to become the real parameters of the model. Also, note that it is equivalent to pose the $\ell_0$ constraints on $\mathbf{I}_r^H, \mathbf{I}_r^T$ instead of $\boldsymbol{\alpha}_r^T, \boldsymbol{\alpha}_r^H$. Putting these modifications together, we can rewrite the optimization problem as

$$\begin{aligned} \text{minimize} \quad & \mathcal{L} \\ \text{subject to} \quad & \|\mathbf{I}_r^H\|_0 \le k, \|\mathbf{I}_r^T\|_0 \le k \end{aligned} \quad (3)$$

where $\mathcal{L}$ is the loss function defined in Eq. (2).

## 3.2 Block Iterative Optimization

Though sparseness is favorable in practice, it is generally NP-hard to find the optimal solution under $\ell_0$ constraints. Thus, we resort to an approximated algorithm in this work.

For convenience, we refer to the parameters with and without the sparse constraints as the *sparse* partition and the *dense* partition, respectively. Based on this notion, the high-level idea of the approximated algorithm is to iteratively optimize one of the two partitions while holding the other one fixed. Since all parameters in the dense partition, including the embeddings, the projection matrices, and the pre-softmax scores, are fully differentiable with the sparse partition fixed, we can simply utilize SGD to optimize the dense partition. Then, the core difficulty lies in the step of optimizing the sparse partition (i.e. the sparse assignment vectors), during which we want the following two properties to hold

1. the sparsity required by the $\ell_0$ constraint is maintained, and
2. the cost define by Eq. (2) is decreased.

Satisfying the two criterion seems to highly resemble the original problem defined in Eq. (3). However, the dramatic difference here is that with parameters in the dense partition regarded as constant, the cost function is decoupled w.r.t. each relation $r$. In other words, the optimal choice of $\mathbf{I}_r^H, \mathbf{I}_r^T$ is independent of $\mathbf{I}_{r'}^H, \mathbf{I}_{r'}^T$ for any $r' \neq r$. Therefore, we only need to consider the optimization for a single relation $r$, which is essentially an assignment problem. Note that, however, $\mathbf{I}_r^H$ and $\mathbf{I}_r^T$ are still coupled, without which we basically reach the situation in a backpack problem. In principle, one can explore combinatorial optimization techniques to optimize $\mathbf{I}_{r'}^H, \mathbf{I}_{r'}^T$ jointly, which usually involve some iterative procedure. To avoid adding another inner loop to our algorithm, we turn to a simple but fast approximation method based on the following single-matrix cost.

Specifically, for each relation $r$, we consider the induced cost $\mathcal{L}_{r,i}^H$ where only a single projection matrix $i$ is used for the head entity:

$$\mathcal{L}_{r,i}^H = \sum_{\substack{(h,r,t)\sim P_r, \\ (h',r,t')\sim N_r}} \left[ \gamma + f_{r,i}^H(h,t) - f_{r,i}^H(h',t') \right]_+$$

where $f_{r,i}^H(h,t) = \|\mathbf{D}_i \cdot \mathbf{h} + \mathbf{r} - \boldsymbol{\alpha}_r^T \cdot \mathbf{D} \cdot \mathbf{t}\|$ is the corresponding energy function, and the subscript in $P_r$ and $N_r$ denotes the subsets with relation $r$. Intuitively, $\mathcal{L}_{r,i}^H$ measures, given the current tail attention vector $\boldsymbol{\alpha}_r^T$, if only one project matrix could be chosen for the head entity, how implausible $D_i$ would be. Hence, $i^* = \arg\min_i \mathcal{L}_{r,i}^H$ gives

us the best single projection matrix on the head side given $\boldsymbol{\alpha}_r^T$.

Now, in order to choose the best $k$ matrices, we basically ignore the interaction among projection matrices, and update $\mathbf{I}_r^H$ in the following way:

$$\mathbf{I}_{r,i}^H \leftarrow \begin{cases} 1, & i \in \operatorname{argpartition}_i(\mathcal{L}_{r,i}^H, k) \\ 0, & \text{otherwise} \end{cases}$$

where the function $\operatorname{argpartition}_i(x_i, k)$ produces the index set of the lowest-$k$ values of $x_i$.

Analogously, we can define the single-matrix cost $\mathcal{L}_{r,i}^T$ and the energy function $f_{r,i}^T(h,t)$ on the tail side in a symmetric way. Then, the update rule for $\mathbf{I}_r^H$ follows the same derivation. Admittedly, the approximation described here is relatively crude. But as we will show in section 4, the proposed algorithm yields good performance empirically. We leave the further improvement of the optimization method as future work.

### 3.3 Corrupted Sample Generating Method

Recall that we need to sample a negative triple $(h', r, t')$ to compute hinge loss shown in Eq. 2, given a positive triple $(h, r, t) \in P$. The distribution of negative triple is denoted by $N(h, r, t)$. Previous work (Bordes et al., 2013; Lin et al., 2015b; Yang et al., 2015; Nguyen et al., 2016b) generally constructs a set of corrupted triples by replacing the head entity or tail entity with a random entity uniformly sampled from the KB.

However, uniformly sampling corrupted entities may not be optimal. Often, the head and tail entities associated a relation can only belong to a specific domain. When the corrupted entity comes from other domains, it is very easy for the model to induce a large energy gap between true triple and corrupted one. As the energy gap exceeds $\gamma$, there will be no training signal from this corrupted triple. In comparison, if the corrupted entity comes from the same domain, the task becomes harder for the model, leading to more consistent training signal.

Motivated by this observation, we propose to sample corrupted head or tail from entities in the same domain with a probability $p_r$ and from the whole entity set with probability $1 - p_r$. The choice of relation-dependent probability $p_r$ is specified in Appendix A.1. In the rest of the paper, we refer to the new proposed sampling method as "domain sampling".

## 4 Experiments

### 4.1 Setup

To evaluate link prediction, we conduct experiments on the WN18 (WordNet) and FB15k (Freebase) introduced by Bordes et al. (2013) and use the same training/validation/test split as in (Bordes et al., 2013). The information of the two datasets is given in Table 1.

| Dataset | #E | #R | #Train | #Valid | #Test |
|---|---|---|---|---|---|
| WN18 | 40,943 | 18 | 141,442 | 5,000 | 5,000 |
| FB15k | 14,951 | 1,345 | 483,142 | 50,000 | 59,071 |

Table 1: Statistics of FB15k and WN18 used in experiments. #E, #R denote the number of entities and relation types respectively. #Train, #Valid and #Test are the numbers of triples in the training, validation and test sets respectively.

In knowledge base completion task, we evaluate model's performance of predicting the head entity or the tail entity given the relation and the other entity. For example, to predict head given relation $r$ and tail $t$ in triple $(h, r, t)$, we compute the energy function $f_r(h', t)$ for each entity $h'$ in the knowledge base and rank all the entities according to the energy. We follow Bordes et al. (2013) to report the *filter* results, i.e., removing all other correct candidates $h'$ in ranking. The rank of the correct entity is then obtained and we report the mean rank (mean of the predicted ranks) and Hits@10 (top 10 accuracy). Lower mean rank or higher Hits@10 mean better performance.

### 4.2 Implementation Details

We initialize the projection matrices with identity matrices added with a small noise sampled from normal distribution $\mathcal{N}(0, 0.005^2)$. The entity and relation vectors of ITransF are initialized by TransE (Bordes et al., 2013), following Lin et al. (2015b); Ji et al. (2015); García-Durán et al. (2016, 2015); Lin et al. (2015a). We ran mini-batch SGD until convergence. We employ the "*Bernoulli*" sampling method to generate incorrect triples as used in Wang et al. (2014), Lin et al. (2015b), He et al. (2015), Ji et al. (2015) and Lin et al. (2015a). We set the

margin $\gamma$ to 5 and dimension of embedding $n$ to 50 for WN18, and $\gamma = 1, n = 100$ for FB15k. We set the batch size to 20 for WN18 and 1000 for FB15k. The learning rate is 0.01 on WN18 and 0.1 on FB15k. We use 30 matrices on WN18 and 300 matrices on FB15k. All the models are implemented with Theano (Bergstra et al., 2010). The Softmax temperature is set to $1/4$.

### 4.3 Results & Analysis

The overall link prediction results[1] are reported in Table 2. Our model consistently outperforms previous models without external information on both the metrics of WN18 and FB15k. On WN18, we even achieve a much better mean rank with comparable Hits@10 than current state-of-the-art model IRN employing external information.

We can see that path information is very helpful on FB15k and models taking advantage of path information outperform intrinsic models by a significant margin. Indeed, a lot of facts are easier to recover with the help of multi-step inference. For example, if we know Barack Obama is born in Honolulu, a city in the United States, then we easily know the nationality of Obama is the United States. An straightforward way of extending our proposed model to $k$-step path $P = \{r_i\}_{i=1}^k$ is to define a path energy function $\|\boldsymbol{\alpha}_P^H \cdot \mathbf{D} \cdot \mathbf{h} + \sum_{r_i \in P} \mathbf{r}_i - \boldsymbol{\alpha}_P^T \cdot \mathbf{D} \cdot \mathbf{t}\|_\ell$, $\boldsymbol{\alpha}_P^H$ is a concept association related to the path. We plan to extend our model to multi-step path in the future.

To provide a detailed understanding why the proposed model achieves better performance, we present some further analysis in the sequel.

**Performance on Rare Relations** In the proposed ITransF, we design an attention mechanism to encourage knowledge sharing across different relations. Naturally, facts associated with rare relations should benefit most from such sharing, boosting the overall performance. To verify this hypothesis, we investigate our model's performance on relations with different frequency. Specifically, we sort all relations by their frequency in the training set, and split them into 5 buckets evenly. Within each bucket, we compare our model with STransE, as shown in Fig-

---

[1]Note that although IRN (Shen et al., 2016) does not explicitly exploit path information, it performs multi-step inference through the multiple usages of external memory. When IRN is allowed to access memory once for each prediction, its Hits@10 is 80.7, similar to models without path information.

| Model | Additional Information | WN18 | | FB15k | |
|---|---|---|---|---|---|
| | | Mean Rank | Hits@10 | Mean Rank | Hits@10 |
| SE (Bordes et al., 2011) | No | 985 | 80.5 | 162 | 39.8 |
| Unstructured (Bordes et al., 2014) | No | 304 | 38.2 | 979 | 6.3 |
| TransE (Bordes et al., 2013) | No | 251 | 89.2 | 125 | 47.1 |
| TransH (Wang et al., 2014) | No | 303 | 86.7 | 87 | 64.4 |
| TransR (Lin et al., 2015b) | No | 225 | 92.0 | 77 | 68.7 |
| CTransR (Lin et al., 2015b) | No | 218 | 92.3 | 75 | 70.2 |
| KG2E (He et al., 2015) | No | 348 | 93.2 | 59 | 74.0 |
| TransD (Ji et al., 2015) | No | 212 | 92.2 | 91 | 77.3 |
| TATEC (García-Durán et al., 2016) | No | - | - | **58** | 76.7 |
| NTN (Socher et al., 2013) | No | - | 66.1 | - | 41.4 |
| DISTMULT (Yang et al., 2015) | No | - | 94.2 | - | 57.7 |
| STransE (Nguyen et al., 2016b) | No | 206 (244) | 93.4 (94.7) | 69 | 79.7 |
| ITransF | No | **205** | 94.2 | 65 | 81.0 |
| ITransF (domain sampling) | No | 223 | **95.2** | 77 | **81.4** |
| RTransE (García-Durán et al., 2015) | Path | - | - | 50 | 76.2 |
| PTransE (Lin et al., 2015a) | Path | - | - | 58 | 84.6 |
| NLFeat (Toutanova and Chen, 2015) | Node + Link Features | - | 94.3 | - | 87.0 |
| Random Walk (Wei et al., 2016) | Path | - | 94.8 | - | 74.7 |
| IRN (Shen et al., 2016) | External Memory | 249 | *95.3* | *38* | 92.7 |

Table 2: Link prediction results on two datasets. Higher Hits@10 or lower Mean Rank indicates better performance. Following Nguyen et al. (2016b) and Shen et al. (2016), we divide the models into two groups. The first group contains intrinsic models without using extra information. The second group make use of additional information. Results in the brackets are another set of results STransE reported.

ure 1.[2] As we can see, on WN18, ITransF outperforms STransE by a significant margin on rare relations. In particular, in the last bin (rarest relations), the average Hits@10 increases from 55.2 to 93.8, showing the great benefits of transferring statistical strength from common relations to rare ones. On FB15k, we can also observe a similar pattern, although the degree of improvement is less significant. We conjecture the difference roots in the fact that many rare relations on FB15k have disjoint domains, knowledge transfer through common concepts is harder.

**Interpretability** In addition to the quantitative evidence supporting the effectiveness of knowledge sharing, we provide some intuitive examples to show how knowledge is shared in our model. As we mentioned earlier, the sparse attention vectors fully capture the association between relations and concepts and hence the knowledge transfer among relations. Thus, we visualize the attention vectors for several relations on both WN18 and FB15K in Figure 2.

For WN18, the words "hyponym" and "hypernym" refer to words with more specific or general meaning respectively. For example, PhD is a hyponym of student and student is a hypernym of PhD. As we can see, concepts associated with the head entities in one relation are also associated

with the tail entities in its reverse relation. Further, "instance_hypernym" is a special hypernym with the head entity being an instance, and the tail entity being an abstract notion. A typical example is (*New York*, instance_hypernym, *city*). This connection has also been discovered by our model, indicated by the fact that "instance_hypernym(T)" and "hypernym(T)" share a common concept matrix. Finally, for symmetric relations like "similar_to", we see the head attention is identical to the tail attention, which well matches our intuition.

On FB15k, we also see the sharing between reverse relations, as in "(somebody) won_award_for (some work)" and "(some work) award_winning_work (somebody)". What's more, although relation "won_award_for" and "was_nominated_for" share the same concepts, their attention distributions are different, suggesting distinct emphasis. Finally, symmetric relations like spouse behave similarly as mentioned before.

**Model Compression** A byproduct of parameter sharing mechanism employed by ITransF is a much more compact model with equal performance. Figure 3 plots the average performance of ITransF against the number of projection matrices $m$, together with two baseline models. On FB15k, when we reduce the number of matrices from 2200 to 30 ($\sim 90\times$ compression), our model performance decreases by only 0.09% on

---
[2]Domain sampling is not employed.

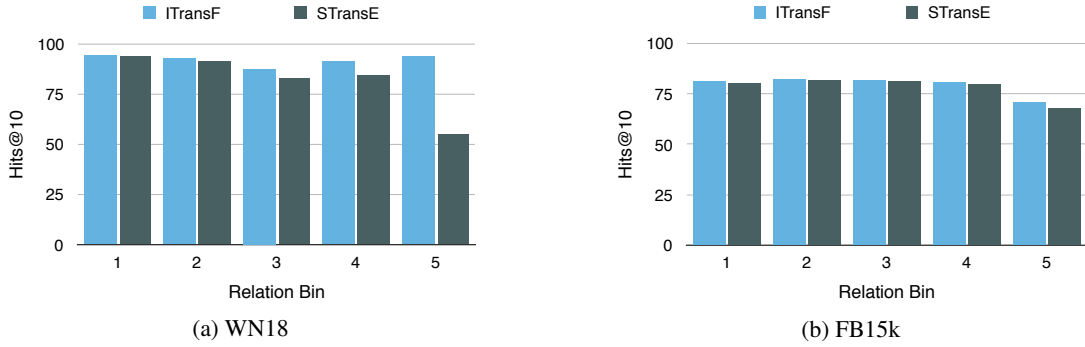

Figure 1: Hits@10 on relations with different amount of data. We give each relation the equal weight and report the average Hits@10 and MR of each relation in a bin instead of reporting the average Hits@10 and MR of each sample in a bin. Bins with smaller index corresponding to high-frequency relations.

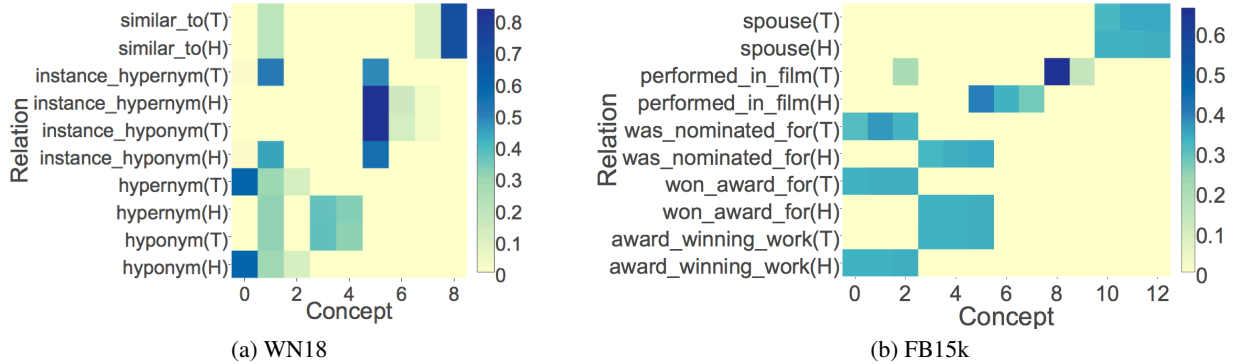

Figure 2: Heatmap visualization of attention vectors on WN18 and FB15k. Each row is an attention vector $\boldsymbol{\alpha}_r^H$ or $\boldsymbol{\alpha}_r^T$ for a relation's head or tail concepts.

Hits@10, still outperforming STransE. Similarly, on WN18, ITransF continues to achieve the best performance when we reduce the number of concept project matrices to 18.

## 5 Analysis on Sparseness

Sparseness is desirable since it contribute to interpretability and computational efficiency of our model. We investigate whether enforcing sparseness would deteriorate the model performance and compare our method with another sparse encoding methods in this section.

**Dense Attention vs Sparse Attention** Although sparsity usually enjoys many practical advantages, it may deteriorate the model performance when applied improperly. Here, we show that our model employing sparse attention can achieve similar results with dense attention with a significantly less computational burden. We compare models in a setting where the computation time of dense attention model is acceptable[3]. We use 22 weight

matrices on WN18 and 15 weight matrices on FB15k and train both the models for 1500 epochs. The results are reported in table 3. Generally, ITransF with sparse attention has slightly better performance comparing to dense attention except a higher mean rank on WN18.

| Method | WN18 | | | FB15k | | |
|--------|------|-----|------|-------|------|------|
| | MR | H10 | Time | MR | H10 | Time |
| Dense | **192** | 93.6 | 4m34s | 69 | 79.2 | 4m30s |
| Sparse | 199 | **93.8** | **2m32s** | **68** | **79.3** | **1m52s** |

Table 3: Performance of model with dense attention vectors or sparse attention vectors. MR, H10 and Time denotes mean rank, Hits@10 and training time per epoch respectively

**Nonnegative Sparse Encoding** In the proposed model, we induce the sparsity by a carefully designed iterative optimization procedure. Apart from this approach, one may utilize sparse encoding techniques to obtain sparseness based on the pretrained projection matrices from STransE. Concretely, stacking $|2R|$ pretrained projection matrices into a 3-dimensional tensor $X \in$

---

[3]With 300 projection matrices, it takes $1h1m$ to run one epoch for a model with dense attention.

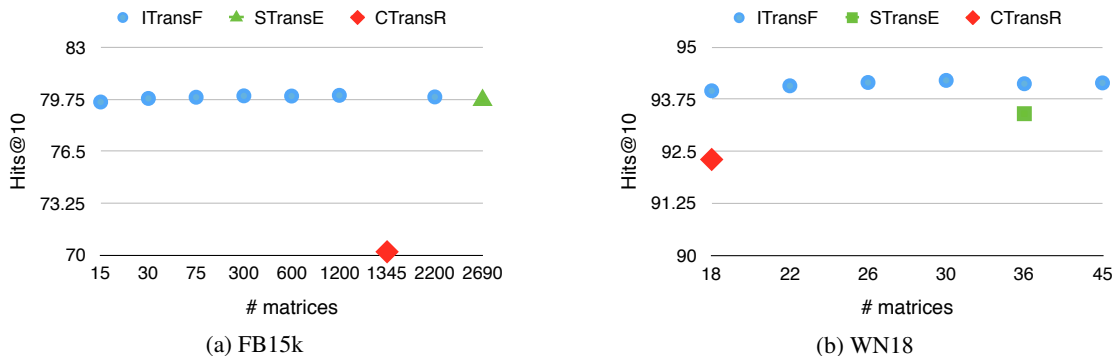

Figure 3: Performance with different number of projection matrices. Note that the X-axis denoting the number of matrices is not linearly scaled.

$\mathbb{R}^{2|R| \times n \times n}$, similar sparsity can be induced by solving an $\ell_1$-regularized tensor completion problem $\min_{\mathbf{A,D}} ||\mathbf{X} - \mathbf{DA}||_2^2 + \lambda ||\mathbf{A}||_{\ell_1}$. Basically, $\mathbf{A}$ plays the same role as the attention vectors in our model. For more details, we refer readers to (Faruqui et al., 2015).

For completeness, we compare our model with the aforementioned approach[4]. The comparison is summarized in table 4. On both benchmarks, ITransF achieves significant improvement against sparse encoding on pretrained model. This performance gap should be expected since the objective function of sparse encoding methods is to minimize the reconstruction loss rather than optimize the criterion for link prediction.

| Method | WN18 | | FB15k | |
|---|---|---|---|---|
| | MR | H10 | MR | H10 |
| Sparse Encoding | 211 | 86.6 | 66 | 79.1 |
| ITransF | **205** | **94.2** | **65** | **81.0** |

Table 4: Different methods to obtain sparse representations

# 6 Related Work

In KBC, CTransR (Lin et al., 2015b) enables relation embedding sharing across similar relations, but they cluster relations before training rather than learning it in a principled way. Further, they do not solve the data sparsity problem because there is no sharing of projection matrices which have a lot more parameters.

Data sparsity is a common problem in many fields. Transfer learning (Pan and Yang, 2010) has been shown to be promising to transfer knowledge and statistical strengths across similar models or languages. For example, Bharadwaj et al.

(2016) transfers models on resource-rich languages to low resource languages by parameter sharing through common phonological features in name entity recognition. Zoph et al. (2016) initialize from models trained by resource-rich languages to translate low-resource languages.

Several works on obtaining a sparse attention (Martins and Astudillo, 2016; Makhzani and Frey, 2014; Shazeer et al., 2017) share a similar idea of sorting the values before softmax and only keeping the $K$ largest values. However, the sorting operation in these works is not GPU-friendly.

The block iterative optimization algorithm in our work is inspired by LightRNN (Li et al., 2016). They allocate every word in the vocabulary in a table. A word is represented by a row vector and a column vector depending on its position in the table. They iteratively optimize embeddings and allocation of words in tables.

# 7 Conclusion and Future Work

In summary, we propose a knowledge embedding model which can discover shared hidden concepts, and design a learning algorithm to induce the interpretable sparse representation. Empirically, we show our model can improve the performance on two benchmark datasets without external resources, over all previous models of the same kind.

In the future, we plan to enable ITransF to perform multi-step inference, and extend the sharing mechanism to entity and relation embeddings, further enhancing the statistical binding across parameters. In addition, our framework can also be applied to multi-task learning, promoting a finer sharing among different tasks.

---

[4] We use the toolkit provided by (Faruqui et al., 2015).

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
