# Peer review of "An Interpretable Knowledge Transfer Model for Knowledge Base Completion"

_ACL 2017 — decision unknown_

[Official Review · Reviewer 1 · rating 3 · confidence 5]
soundness 5 · originality 3 · clarity 4 · impact 3 · substance 3 · appropriateness 5 · meaningful comparison 4 · presentation format Poster

This paper considers the problem of KB completion and proposes ITransF for this
purpose. Unlike STransE that assigns each relation an independent matrix, this
paper proposes to share the parameters between different relations. A model is
proposed where a tensor D is constructed that contains various relational
matrices as its slices and a selectional vector \alpha is used to select a
subset of relevant relational matrix for composing a particular semantic
relation. The paper then discuss a technique to make \alpha sparse.
Experimental results on two standard benchmark datasets shows the superiority
of ITransF over prior proposals.

The paper is overall well written and the experimental results are good.
However, I have several concerns regarding this work that I hope the authors
will answer in their response.

1. Just by arranging relational matrices in a tensor and selecting (or more
appropriately considering a linearly weighted sum of the relational matrices)
does not ensure any information sharing between different relational matrices.
This would have been the case if you had performed some of a tensor
decomposition and projected the different slices (relational matrices) into
some common lower-dimensional core tensor. It is not clear why this approach
was not taken despite the motivation to share information between different
relational matrices.
2. The two requirements (a) to share information across different relational
matrices and (b) make the attention vectors sparse are some what contradictory.
If the attention vector is truly sparse and has many zeros then information
will not flow into those slices during optimisation. 
3. The authors spend a lot of space discussing techniques for computing sparse
attention vectors. The authors mention in page 3 that \ell_1 regularisation did
not work in their preliminary experiments. However, no experimental results are
shown for \ell_1 regularisation nor they explain why \ell_1 is not suitable for
this task. To this reviewer, it appears as an obvious baseline to try,
especially given the ease of optimisation. You use \ell_0 instead and get into
NP hard optimisations because of it. Then you propose a technique and a rather
crude approximation in the end to solve it. All that trouble could be spared if
\ell_1 was used.
4. The vector \alpha is performing a selection or a weighing over the slices of
D. It is slightly misleading to call this as “attention” as it is a term
used in NLP for a more different type of models (see attention model used in
machine translation).
5. It is not clear why you need to initialise the optimisation by pre-trained
embeddings from TransE. Why cannot you simply randomly initialise the
embeddings as done in TransE and then update them? It is not fair to compare
against TransE if you use TransE as your initial point.

Learning the association between semantic relations is an idea that has been
used in related problems in NLP such as relational similarity measurement
[Turney JAIR 2012] and relation adaptation [Bollegala et al. IJCAI 2011]. It
would be good to put the current work with respect to such prior proposals in
NLP for modelling inter-relational correlation/similarity.

Thanks for providing feedback. I have read it.